# Layer-Dependent Importance Sampling for Training Deep and Large Graph Convolutional Networks

**Difan Zou**\*, **Ziniu Hu**\*, **Yewen Wang, Song Jiang, Yizhou Sun, Quanquan Gu**
Department of Computer Science, UCLA, Los Angeles, CA 90095
{knowzou,bull,wyw10804,songjiang,yzsun,qgu}@cs.ucla.edu

## Abstract

Graph convolutional networks (GCNs) have recently received wide attentions, due to their successful applications in different graph tasks and different domains. Training GCNs for a large graph, however, is still a challenge. Original full-batch GCN training requires calculating the representation of all the nodes in the graph per GCN layer, which brings in high computation and memory costs. To alleviate this issue, several sampling-based methods have been proposed to train GCNs on a subset of nodes. Among them, the node-wise neighbor-sampling method recursively samples a fixed number of neighbor nodes, and thus its computation cost suffers from exponential growing neighbor size; while the layer-wise importance-sampling method discards the neighbor-dependent constraints, and thus the nodes sampled across layer suffer from sparse connection problem. To deal with the above two problems, we propose a new effective sampling algorithm called LAyer-Dependent ImportancE Sampling (LADIES) [2]. Based on the sampled nodes in the upper layer, LADIES selects their neighborhood nodes, constructs a bipartite subgraph and computes the importance probability accordingly. Then, it samples a fixed number of nodes by the calculated probability, and recursively conducts such procedure per layer to construct the whole computation graph. We prove theoretically and experimentally, that our proposed sampling algorithm outperforms the previous sampling methods in terms of both time and memory costs. Furthermore, LADIES is shown to have better generalization accuracy than original full-batch GCN, due to its stochastic nature.

## 1 Introduction

Graph convolutional networks (GCNs) recently proposed by Kipf et al. [12] adopt the concept of convolution filter into graph domain [2, 6–8]. For a given node, a GCN layer aggregates the embeddings of its neighbors from the previous layer, followed by a non-linear transformation, to obtain an updated contextualized node representation. Similar to the convolutional neural networks (CNNs) [13] in the computer vision domain, by stacking multiple GCN layers, each node representation can utilize a wide receptive field from both its immediate and distant neighbors, which intuitively increases the model capacity.

Despite the success of GCNs in many graph-related applications [12, 17, 15], training a deep GCN for large-scale graphs remains a big challenge. Unlike tokens in a paragraph or pixels in an image, which normally have limited length or size, graph data in practice can be extremely large. For example, Facebook social network in 2019 contains 2.7 billion users[3]. Such a large-scale graph is impossible to be handled using full-batch GCN training, which takes all the nodes into one batch to update parameters. However, conducting mini-batch GCN training is non-trivial, as the nodes

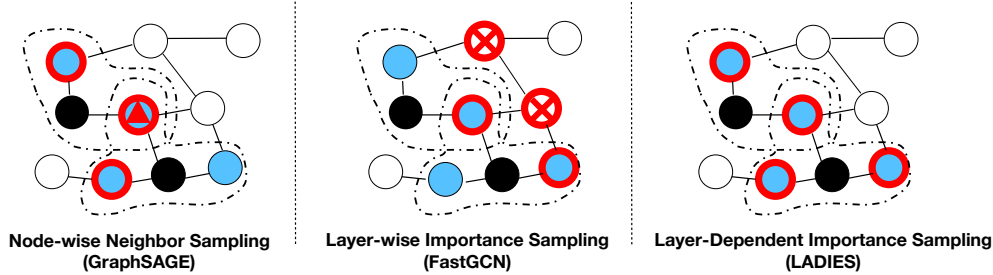

| Node-wise Neighbor Sampling (GraphSAGE) | Layer-wise Importance Sampling (FastGCN) | Layer-Dependent Importance Sampling (LADIES) |

Figure 1: An illustration of the sampling process of GraphSage, FastGCN, and our proposed LADIES. Black nodes denote the nodes in the upper layer, blue nodes in the dashed circle are their neighbors, and node with the red frame is the sampled nodes. As is shown in the figure, GraphSAGE will redundantly sample a neighboring node twice, denoted by the red triangle, while FastGCN will sample nodes outside of the neighborhood. Our proposed LADIES can avoid these two problems.

in a graph are closely coupled and correlated. In particular, in GCNs, the embedding of a given node depends recursively on all its neighbors' embeddings, and such computation dependency grows exponentially with respect to the number of layers. Therefore, training deep and large GCNs is still a challenge, which prevents their application to many large-scale graphs, such as social networks [12], recommender systems [17], and knowledge graphs [15].

To alleviate the previous issues, researchers have proposed sampling-based methods to train GCNs based on mini-batch of nodes, which only aggregate the embeddings of a sampled subset of neighbors of each node in the mini-batch. Among them, one direction is to use a node-wise neighbor-sampling method. For example, GraphSAGE [9] calculates each node embedding by leveraging only a fixed number of uniformly sampled neighbors. Although this kind of approaches reduces the computation cost in each aggregation operation, the total cost can still be large. As is pointed out in [11], the recursive nature of node-wise sampling brings in redundancy for calculating embeddings. Even if two nodes share the same sampled neighbor, the embedding of this neighbor has to be calculated twice. Such redundant calculation will be exaggerated exponentially when the number of layers increases. Following this line of research as well as reducing the computation redundancy, a series of work was proposed to reduce the size of sampled neighbors. VR-GCN [3] proposes to leverage variance reduction techniques to improve the sample complexity. Cluster-GCN [5] considers restricting the sampled neighbors within some dense subgraphs, which are identified by a graph clustering algorithm before the training of GCN. However, these methods still cannot well address the issue of redundant computations, which may become worse when training very deep and large GCNs.

Another direction uses a layer-wise importance-sampling method. For example, FastGCN [4] calculates a sampling probability based on the degree of each node, and samples a fixed number of nodes for each layer accordingly. Then, it only uses the sampled nodes to build a much smaller sampled adjacency matrix, and thus the computation cost is reduced. Ideally, the sampling probability is calculated to reduce the estimation variance in FastGCN [4], and guarantee fast and stable convergence. However, since the sampling probability is independent for each layer, the sampled nodes from two consecutive layers are not necessarily connected. Therefore, the sampled adjacency matrix can be extremely sparse, and may even have all-zero rows, meaning some nodes are disconnected. Such a sparse connection problem incurs an inaccurate computation graph and further deteriorates the training and generalization performance of FastGCN. In order to capture the inter-layer correlation and reduce the estimation variance, Huang et al. [11] proposed an adaptive and trainable sampling method that conducts layer-wise sampling conditioned on the former layer, which has been demonstrated to achieve higher accuracy than FastGCN. Yet the advantage of the importance sampling procedure used in [11] in terms of time and memory costs is not fully justified theoretically or empirically.

Based on the pros and cons of the aforementioned sampling approaches, we argue that an ideal sampling method should have the following features: 1) *layer-wise*, thus the neighbor nodes can be taken into account together to calculate next layers' embeddings without redundancy; 2) *neighbor-dependent*, thus the sampled adjacency matrix is dense without losing much information for training; 3) the *importance sampling* method should be adopted to reduce the sampling variance and accelerate

convergence. To this end, we propose a new efficient sampling algorithm called LAyer-Dependent ImportancE-Sampling (LADIES) [4], which fulfills all the above features.

The procedure of LADIES is described as below: (1) For each current layer ($l$), based on the nodes sampled in the upper layer ($l + 1$), it picks all their neighbors in the graph, and constructs a bipartite graph among the nodes between the two layers. (2) Then it calculates the sampling probability according to the degree of nodes in the current layer, with the purpose to reduce sampling variance. (3) Next, it samples a fixed number of nodes based on this probability. (4) Finally, it constructs the sampled adjacency matrix between layers and conducts training and inference, where row-wise normalization is applied to all sampled adjacency matrices to stabilize training. As illustrated in Figure 1, our proposed sampling strategy can avoid two pitfalls faced by existing two sampling strategies: layer-wise structure avoids exponential expansion of receptive field; layer-dependent importance sampling guarantees the sampled adjacency matrix to be dense such that the connectivity between nodes in two adjacent layers can be well maintained.

We highlight our contributions as follows:

- We propose LAyer-Dependent ImportancE Sampling (LADIES) for training deep and large graph convolutional networks, which is built upon a novel layer-dependent sampling scheme to avoid exponential expansion of receptive field as well as guarantee the connectivity of the sampled adjacency matrix.
- We prove theoretically that the proposed algorithm achieves significantly better memory and time complexities compared with node-wise sampling methods including GraphSage [9] and VR-GCN [3], and has a dramatically smaller variance compared with FastGCN [4].
- We evaluate the performance of the proposed LADIES algorithm on benchmark datasets and demonstrate its superior performance in terms of both running time and test accuracy. Moreover, we show that LADIES achieves high accuracy with an extremely small sample size (e.g., 256 for a graph with 233k nodes), which enables the use of LADIES for training very deep and large GCNs.

## 2 Background

In this section, we review GCNs and several state-of-the-art sampling-based training algorithms, including full-batch, node-wise neighbor sampling methods, and layer-wise importance sampling methods. We summarize the notation used in this paper in Table 1.

### 2.1 Existing GCN Training Algorithms

**Full-batch GCN.** When given an undirected graph $\mathcal{G}$, with $\mathbf{P}$ defined in Table 1, the $l$-th GCN layer is defined as:

$$\mathbf{Z}^{(l)} = \mathbf{P}\mathbf{H}^{(l-1)}\mathbf{W}^{(l-1)}, \quad \mathbf{H}^{(l-1)} = \sigma(\mathbf{Z}^{(l-1)}), \tag{1}$$

Considering a node-level downstream task, given training dataset $\{(\mathbf{x}_i, y_i)\}_{v_i \in \mathcal{V}_s}$, the weight matrices $\mathbf{W}^{(l)}$ will be learned by minimizing the loss function: $\mathcal{L} = \frac{1}{|\mathcal{V}_s|} \sum_{v_i \in \mathcal{V}_s} \ell(y_i, \mathbf{z}_i^{(L)})$, where $\ell(\cdot, \cdot)$ is a specified loss function, $\mathbf{z}_i^L$ denotes the output of GCN corresponding to the vertex $v_i$, and $\mathcal{V}_S$ denotes the training node set. Gradient descent algorithm is utilized for the full-batch optimization, where the gradient is computed over all the nodes as $\frac{1}{|\mathcal{V}_s|} \sum_{v_i \in \mathcal{V}_S} \nabla \ell(y_i, \mathbf{z}_i^{(L)})$.

When the graph is large and dense, full-batch GCN's memory and time costs can be very expensive while computing such gradient, since during both backward and forward propagation process it requires to store and aggregate embeddings for all nodes across all layer. Also, since each epoch only updates parameters once, the convergence would be very slow.

One solution to address issues of full-batch training is to sample a a mini-batch of labeled nodes $\mathcal{V}_B \in \mathcal{V}_S$, and compute the mini-batch stochastic gradient $\frac{1}{|\mathcal{V}_B|} \sum_{v_i \in \mathcal{V}_B} \nabla \ell(y_i, \mathbf{z}_i^L)$. This can help reduce computation cost to some extent, but to calculate the representation of each output node, we still need to consider a large-receptive field, due to the dependency of the nodes in the graph.

**Node-wise Neighbor Sampling Algorithms.** GraphSAGE [9] proposed to reduce receptive field size by neighbor sampling (NS). For each node in $l$-th GCN layer, NS randomly samples $s_{node}$

Table 1: Summary of Notations

| | |
|---|---|
| $\mathcal{G} = (\mathcal{V}, \mathbf{A}), \|\mathbf{A}\|_0$ | $\mathcal{G}$ denotes a graph consist of a set of nodes $\mathcal{V}$ with node number $|\mathcal{V}|$, $\mathbf{A}$ is the adjacency matrix, and $\|\mathbf{A}\|_0$ denotes the number of non-zero entries in $\mathbf{A}$. |
| $\mathbf{M}_{i,*}, \mathbf{M}_{*,j}\ \mathbf{M}_{i,j}$ | $\mathbf{M}_{i,*}$ is the i-th row of matrix M, $\mathbf{M}_{*,j}$ is the j-th column of matrix M, and $\mathbf{M}_{i,j}$ is the entry at the position $(i,j)$ of matrix M. |
| $\tilde{\mathbf{A}}, \tilde{\mathbf{D}}, \mathbf{P}$ | $\tilde{\mathbf{A}} = \mathbf{A} + \mathbf{I}$, $\tilde{\mathbf{D}}$ is a diagonal matrix satisfying $\tilde{\mathbf{D}}_{i,i} = \sum_j \tilde{\mathbf{A}}_{i,j}$, and $\mathbf{P} = \tilde{\mathbf{D}}^{-\frac{1}{2}}\tilde{\mathbf{A}}\tilde{\mathbf{D}}^{-\frac{1}{2}}$ is the normalized Laplacian matrix. |
| $l, \sigma(\cdot), \mathbf{H}^{(l)}, \mathbf{Z}^{(l)}, \mathbf{W}^{(l)}$ | $l$ denotes the GCN layer index, $\sigma(\cdot)$ denotes the activation function, $\mathbf{H}^{(l)} = \sigma(\mathbf{Z}^{(l)})$ denotes the embedding matrix at layer $l$, $\mathbf{H}^{(0)} = \mathbf{X}$, $\mathbf{Z}^{(l)} = \mathbf{P}\mathbf{H}^{(l-1)}\mathbf{W}^{(l-1)}$ is the intermediate embedding matrix, and $\mathbf{W}^{(l)}$ denotes the trainable weight matrix at layer $l$. |
| $L, K$ | $L$ is the total number of layers in GCN, and $K$ is the dimension of embedding vectors (for simplicity, assume it is the same across all layers). |
| $b, s_{node}, s_{layer}$ | For batch-wise sampling, $b$ denotes the batch size, $s_{node}$ is the number of sampled neighbors per node for node-wise sampling, and $s_{layer}$ is number of sampled nodes per layer for layer-wise sampling. |

Table 2: Summary of Complexity and Variance [6]. Here $\phi$ denotes the upper bound of the $\ell_2$ norm of embedding vector, $\Delta\phi$ denotes the bound of the norm of the difference between the embedding and its history, $D$ denotes the average degree, $b$ denotes the batch size, and $\bar{V}(b)$ denotes the average number of nodes which are connected to the nodes sampled in the training batch.

| Methods | Memory Complexity | Time Complexity | Variance |
|---|---|---|---|
| Full-Batch [12] | $\mathcal{O}(L|\mathcal{V}|K + LK^2)$ | $\mathcal{O}(L\|\mathbf{A}\|_0 K + L|\mathcal{V}|K^2)$ | $0$ |
| GraphSage [9] | $\mathcal{O}(bKs_{node}^{L-1} + LK^2)$ | $\mathcal{O}(bKs_{node}^{L} + bK^2 s_{node}^{L-1})$ | $\mathcal{O}(D\phi\|\mathbf{P}\|_F^2/(|\mathcal{V}|s_{node}))$ |
| VR-GCN [3] | $\mathcal{O}(L|\mathcal{V}|K + LK^2)$ | $\mathcal{O}(bDKs_{node}^{L-1} + bK^2 s_{node}^{L-1})$ | $\mathcal{O}(D\Delta\phi\|\mathbf{P}\|_F^2/(|\mathcal{V}|s_{node}))$ |
| FastGCN [4] | $\mathcal{O}(LKs_{layer} + LK^2)$ | $\mathcal{O}(LKs_{layer}^2 + LK^2 s_{layer})$ | $\mathcal{O}(\phi\|\mathbf{P}\|_F^2/s_{layer})$ |
| **LADIES** | $\mathcal{O}(LKs_{layer} + LK^2)$ | $\mathcal{O}(LKs_{layer}^2 + LK^2 s_{layer})$ | $\mathcal{O}(\phi\|\mathbf{P}\|_F^2 \bar{V}(b)/(|\mathcal{V}|s_{layer}))$ |

of its neighbors at the $(l-1)$-th GCN layer and formulate an unbiased estimator $\hat{\mathbf{P}}^{(l-1)}\mathbf{H}^{(l-1)}$ to approximate $\mathbf{P}\mathbf{H}^{(l-1)}$ in graph convolution layer:

$$\hat{\mathbf{P}}_{i,j}^{(l-1)} = \begin{cases} \frac{|\mathcal{N}(v_i)|}{s_{node}}\mathbf{P}_{i,j}, & v_j \in \hat{\mathcal{N}}^{(l-1)}(v_i); \\ 0, & \text{otherwise.} \end{cases} \tag{2}$$

where $\mathcal{N}(v_i)$ and $\hat{\mathcal{N}}^{(l-1)}(v_i)$ are the full and sampled neighbor sets of node $v_i$ for $(l-1)$-th GCN layer respectively.

VR-GCN [3] is another neighbor sampling work. It proposed to utilize historical activations to reduce the variance of the estimator under the same sample strategy as GraphSAGE. Though successfully achieved comparable convergence rate, the memory complexity is higher and the time complexity remains the same. Though NS scheme alleviates the memory bottleneck of GCN, there exists redundant computation under NS since the embedding calculation for each node in the same layer is independent, thus the complexity grows exponentially when the number of layers increases.

**Layer-wise Importance Sampling Algorithms.** FastGCN [4] proposed a more advanced layer-wise importance sampling (IS) scheme aiming to solve the scalability issue as well as reducing the variance. IS conducts sampling for each layer with a degree-based sampling probability. The approximation of the $i$-th row of $(\mathbf{P}\mathbf{H}^{(l-1)})$ with $s_{layer}$ samples $v_{j_1}, \ldots, v_{j_{s_l}}$ per layer can be estimated as:

$$(\mathbf{P}\mathbf{H}^{(l-1)})_{i,*} \simeq \frac{1}{s_{layer}} \sum_{k=1, v_{j_k} \sim q(v)}^{s_{layer}} \mathbf{P}_{i,j_k}\mathbf{H}_{j_k,*}^{(l-1)}/q(v_{j_k}) \tag{3}$$

where $q(v)$ is the distribution over $\mathcal{V}$, $q(v_j) = \|\mathbf{P}_{*,j}\|_2^2/\|\mathbf{P}\|_F^2$ is the probability assigned to node $v_j$.

Though IS outperforms uniform sampling and the layer-wise sampling successfully reduced both time and memory complexities, however, this sampling strategy has a major limitation: since the sampling operation is conducted independently at each layer, FastGCN cannot guarantee connectivity between sampled nodes at different layers, which incurs large variance of the approximate embeddings.

## 2.2 Complexity and Variance Comparison

We compare each method's memory, time complexity, and variance with that of our proposed LADIES algorithm in Table 2.

**Complexity.** We now compare the complexity of the proposed LADIES algorithm and existing sampling-based GCN training algorithms. The complexity for all methods are summarized in Table 2, detailed calculation could be found in Appendix A. Compare with full-batch GCN, the time and memory complexities of LADIES do not depend on the total number of nodes and edges, thus our algorithm does not have scalability issue on large and dense graphs. Unlike NS based methods including GraphSAGE and VR-GCN, LADIES is not sensitive to the number of layers and will not suffer exponential growth in complexity, therefore it can perform well when the neural network goes deeper. Compared to layer-wise importance sampling proposed in FastGCN, it maintains the same complexity while obtaining a better convergence guarantee as analyzed in the next paragraph. In fact, in order to guarantee good performance, our method requires a much smaller sample size than that of FastGCN, thus the time and memory burden is much lighter. Therefore, our proposed LADIES algorithm can achieve the best time and memory complexities and is applicable to training very deep and large graph convolution networks.

**Variance.** We compare the variance of our algorithm with existing sampling-based algorithms. To simplify the analysis, when evaluating the variance we only consider two-layer GCN. The results are summarized in Table 2. We defer the detailed calculation to Appendix B. Compared with FastGCN, our variance result is strictly better since $\bar{V}(b) \leq |\mathcal{V}|$, where $\bar{V}(b)$ denotes the number of nodes which are connected to the nodes sampled in the training batch. Moreover, $\bar{V}(b)$ scales with $b$, which implies that our method can be even better when using a small batch size. Compared with node-wise sampling, consider the same sample size, i.e., $s_{layer} = bs_{node}$. Ignoring the same factors, the variance of LADIES is in the order of $\mathcal{O}(\bar{V}(b)/b)$ and the variance of GraphSAGE is $\mathcal{O}(D)$, where $D$ denotes the average degree. Based on the definition of $\bar{V}(b)$, we strictly have $\bar{V}(b) \leq \mathcal{O}(bD)$ since there is no redundant node been calculated in $\bar{V}(b)$. Therefore our method is also strictly better than GraphSAGE especially when the graph is dense, i.e., many neighbors can be shared. The variance of VR-GCN resembles that of GraphSAGE but relies on the difference between the embedding and its history, which is not directly comparable to our results.

# 3 LADIES: LAyer-Dependent ImportancE Sampling

We present our method, LADIES, in this section. As illustrated in previous sections, for node-wise sampling methods [9, 3], one has to sample a certain number of nodes in the neighbor set of all sampled nodes in the current layer, then the number of nodes that are selected to build the computation graph is exponentially large with the number of hidden layers, which further slows down the training process of GCNs. For the existing layer-wise sampling scheme [4], it is inefficient when the graph is sparse, since some nodes may have no neighbor been sampled during the sampling process, which results in meaningless zero activations [3].

To address the aforementioned drawbacks and weaknesses of existing sampling-based methods for GCN training, we propose our training algorithms that can achieve good convergence performance as well as reduce sampling complexity. In the following, we are going to illustrate our method in detail.

## 3.1 Revisiting Independent Layer-wise Sampling

We first revisit the independent layer-wise sampling scheme for building the computation graph of GCN training. Recall in the forward process of GCN (1), the matrix production $\mathbf{PH}^{(l-1)}$ can be regarded as the embedding aggregation process. Then, the layer-wise sampling methods aim to approximate the intermediate embedding matrix $\mathbf{Z}^{(l)}$ by only sampling a subset of nodes at the $(l-1)$-th layer and aggregating their embeddings for approximately estimating the embeddings at the $l$-th layer. Mathematically, similar to (3), let $\mathcal{S}_{(l-1)}$ with $|\mathcal{S}_{l-1}| = s_{l-1}$ be the set of sampled nodes at the $(l-1)$-th layer, we can approximate $\mathbf{PH}^{(l-1)}$ as

$$\mathbf{PH}^{(l-1)} \simeq \frac{1}{s_{l-1}} \sum_{k \in \mathcal{S}^{(l-1)}} \frac{1}{p_k^{(l-1)}} \mathbf{P}_{*,k} \mathbf{H}_{k,*}^{(l-1)},$$

where we adopt non-uniformly sampling scheme by assigning probabilities $p_1^{(l-1)}, \ldots, p_{|\mathcal{V}|}^{(l-1)}$ to all nodes in $\mathcal{V}$. Then the corresponding discount weights are $\{1/(s_{l-1}p_i^{(l-1)})\}_{i=1,\ldots,|\mathcal{V}|}$. Then let

$\{i_k^{(l-1)}\}_{k \in \mathcal{S}_{l-1}}$ be the indices of sampled nodes at the $l-1$-th layer, the estimator of $\mathbf{PH}^{(l-1)}$ can be formulated as

$$\mathbf{PH}^{(l-1)} \simeq \mathbf{PS}^{(l-1)}\mathbf{H}^{(l-1)}, \tag{4}$$

where $\mathbf{S}^{(l-1)} \in \mathbb{R}^{|\mathcal{V}| \times |\mathcal{V}|}$ is a diagonal matrix with only nonzero diagnoal matrix, defined by

$$\mathbf{S}_{s,s}^{(l-1)} = \begin{cases} \frac{1}{s_{l-1}p_{i_k^{(l-1)}}^{(l-1)}}, & s = i_k^{(l-1)}; \\ 0, & \text{otherwise.} \end{cases} \tag{5}$$

It can be verified that $\|\mathbf{S}^{(l-1)}\|_0 = s_{l-1}$ and $\mathbb{E}[\mathbf{S}^{(l-1)}] = \mathbf{I}$. Assuming at the $l$-th and $l-1$-th layers the sets of sampled nodes are determined. Then let $\{i_k^{(l)}\}_{k \in \mathcal{S}_l}$ and $\{i_k^{(l-1)}\}_{k \in \mathcal{S}_{l-1}}$ denote the indices of sampled nodes at these two layers, and define the row selection matrix $\mathbf{Q}^{(l)} \in \mathbb{R}^{s_l \times |\mathcal{V}|}$ as:

$$\mathbf{Q}_{k,s}^{(l)} = \begin{cases} 1 & (k,s) = (k, i_k^{(l)}); \\ 0, & \text{otherwise,} \end{cases} \tag{6}$$

the forward process of GCN with layer-wise sampling can be approximated by

$$\tilde{\mathbf{Z}}^{(l)} = \tilde{\mathbf{P}}^{(l-1)}\tilde{\mathbf{H}}^{(l-1)}\mathbf{W}^{(l-1)}, \quad \tilde{\mathbf{H}}^{(l-1)} = \sigma(\tilde{\mathbf{Z}}^{(l-1)}), \tag{7}$$

where $\tilde{\mathbf{Z}}^{(l)} \in \mathbb{R}^{s_l \times d}$ denotes the approximated intermediate embeddings for sampled nodes at the $l$-th layer and $\tilde{\mathbf{P}}^{(l-1)} = \mathbf{Q}^{(l)}\mathbf{PS}^{(l-1)}\mathbf{Q}^{(l-1)\top} \in \mathbb{R}^{s_l \times s_{l-1}}$ serves as a modified Laplacian matrix, and can be also regarded as a sampled bipartite graph after certain rescaling. Since typically we have $s_l, s_{l-1} \ll |\mathcal{V}|$, the sampled graph is dramatically smaller than the entire one, thus the computation cost can be significantly reduced.

## 3.2 Layer-dependent Importance Sampling

However, independently conducting layer-wise sampling at different layers is not efficient since the sampled bipartite graph may still be sparse and even have all-zero rows. This further results in very poor performance and require us to sample lots of nodes in order to guarantee convergence throughout the GCN training. To alleviate this issue, we propose to apply neighbor-dependent sampling that can leverage the dependency between layers which further leads to dense computation graph. Specifically, our layer-dependent sampling mechanism is designed in a top-down manner, i.e., the sampled nodes at the $l$-th layer are generated depending on the sampled nodes that have been generated in all upper layers. Note that for each node we only need to aggregate the embeddings from its neighbor nodes in the previous layer. Thus, at one particular layer, we only need to generate samples from the union of neighbors of the nodes we have sampled in the upper layer, which is defined by

$$\mathcal{V}^{(l-1)} = \cup_{v_i \in \mathcal{S}_l}\mathcal{N}(v_i)$$

where $\mathcal{S}_l$ is the set of nodes we have sampled at the $l$-th layer and $\mathcal{N}(v_i)$ denotes the neighbors set of node $v_i$. Therefore during the sampling process, we only assign probability to nodes in $\mathcal{V}^{(l-1)}$, denoted by $\{p_i^{(l-1)}\}_{v_i \in \mathcal{V}^{(l-1)}}$. Similar to FastGCN [4], we apply importance sampling to reduce the variance. However, we have no information regarding the activation matrix $\mathbf{H}^{(l-1)}$ when characterizing the samples at the $(l-1)$-th layer. Therefore, we resort to a important sampling scheme which only relies on the matrices $\mathbf{Q}^{(l)}$ and $\mathbf{P}$. Specifically, we define the importance probabilities as:

$$p_i^{(l-1)} = \frac{\|\mathbf{Q}^{(l)}\mathbf{P}_{*,i}\|_2^2}{\|\mathbf{Q}^{(l)}\mathbf{P}\|_F^2}. \tag{8}$$

Evidently, if $v_i \notin \mathcal{V}^{(l-1)}$, we have $\|\mathbf{Q}^{(l)}\mathbf{P}_{*,i}\|_2^2 = 0$, which implies that $p_i^{(l-1)} = 0$. Then, let $\{i_k^{(l-1)}\}_{v_k \in \mathcal{S}_{l-1}}$ be the indices of sampled nodes at the $(l-1)$-th layer based on the importance probabilities computed in (8), we can also define the random diagonal matrix $\mathbf{S}^{(l-1)}$ according to (5), and formulate the same forward process of GCN as in (7) but with a different modified Laplacian matrix $\tilde{\mathbf{P}}^{(l-1)} = \mathbf{Q}^{(l)}\mathbf{PS}^{(l-1)}\mathbf{Q}^{(l-1)\top} \in \mathbb{R}^{s_l \times s_{l-1}}$. The computation of $\tilde{\mathbf{P}}^{(l-1)}$ can be very efficient since it only involves sparse matrix productions. Here the major difference between our sampling

---

**Algorithm 1** Sampling Procedure of LADIES

---

**Require:** Normalized Laplacian Matrix $\mathbf{P}$; Batch Size $b$, Sample Number $n$;
 1: Randomly sample a batch of $b$ output nodes as $\mathbf{Q}^L$
 2: **for** $l = L$ to 1 **do**
 3:　　Get layer-dependent laplacian matrix $\mathbf{Q}^{(l)}\mathbf{P}$. Calculate sampling probability for each node
　　　using $p_i^{(l-1)} \leftarrow \frac{\|\mathbf{Q}^{(l)}\mathbf{P}_{*,i}\|_2^2}{\|\mathbf{Q}^{(l)}\mathbf{P}\|_F^2}$, and organize them into a random diagonal matrix $S^{(l-1)}$.
 4:　　Sample $n$ nodes in $l-1$ layer using $p^{(l-1)}$. The sampled nodes formulate $\mathbf{Q}^{(l-1)}$
 5:　　Reconstruct sampled laplacian matrix between sampled nodes in layer $l-1$ and $l$ by
　　　$\tilde{\mathbf{P}}^{(l-1)} \leftarrow \mathbf{Q}^{(l)}\mathbf{P}S^{(l-1)}\mathbf{Q}^{(l-1)\top}$, then normalize it by $\tilde{\mathbf{P}}^{(l)} \leftarrow \mathbf{D}_{\tilde{\mathbf{P}}^{(l)}}^{-1}\tilde{\mathbf{P}}^{(l)}$.
 6: **end for**
 7: **return** Modified Laplacian Matrices $\{\tilde{\mathbf{P}}^{(l)}\}_{l=1,...,L}$ and Sampled Node at Input Layer $\mathbf{Q}^0$;

---

method and independent layer-wise sampling is the different constructions of matrix $\mathbf{S}^{(l-1)}$. In our sampling mechanism, we have $\mathbb{E}[\mathbf{S}^{(l-1)}] = \mathbf{L}^{(l-1)}$, where $\mathbf{L}^{(l-1)}$ is a diagonal matrix with

$$\mathbf{L}_{s,s}^{(l-1)} = \begin{cases} 1 & s \in \mathcal{V}^{(l-1)} \\ 0, & \text{otherwise.} \end{cases} \tag{9}$$

Since $\|\mathbf{L}^{(l-1)}\|_0 = |\mathcal{V}^{(l-1)}| \ll |\mathcal{V}|$, applying independent sampling method results in the fact that many nodes are in the set $\mathcal{V}/\mathcal{V}^{(l-1)}$, which has no contribution to the construction of computation graph. In contrast, we only sample nodes from $\mathcal{V}^{(l-1)}$ which can guarantee more connections between the sampled nodes at $l$-th and $(l-1)$-th layers, and further leads to a dense computation graph between these two layers.

### 3.3 Normalization

Note that for original GCN, the Laplacian matrix $\mathbf{P}$ is obtained by normalizing the matrix $\mathbf{I} + \mathbf{A}$. Such normalization operation is crucial since it is able to maintain the scale of embeddings in the forward process and avoid exploding/vanishing gradient. However, the modified Laplacian matrix $\{\tilde{\mathbf{P}}^{(l)}\}_{l=1,...,L}$ may not be able to achieve this, especially when $L$ is large, because its maximum singular values can be very large without sufficient samples. Therefore, motivated by [12], we propose to normalize $\tilde{\mathbf{P}}^{(l)}$ such that the sum of all rows are 1, i.e., we have

$$\tilde{\mathbf{P}}^{(l)} \leftarrow \mathbf{D}_{\tilde{\mathbf{P}}^{(l)}}^{-1}\tilde{\mathbf{P}}^{(l)},$$

where $\mathbf{D}_{\tilde{\mathbf{P}}^{(l)}} \in \mathbb{R}^{s_{l+1} \times s_{l+1}}$ is a diagonal matrix with each diagonal entry to be the sum of the corresponding row in $\tilde{\mathbf{P}}^{(l)}$. Now, we can leverage the modified Laplacian matrices $\{\tilde{\mathbf{P}}\}_{l=1,...,L}$ to build the whole computation graph. We formally summarize the proposed algorithm in Algorithm 1.

## 4 Experiments

In this section, we conduct experiments to evaluate LADIES for training deep GCNs on different node classification datasets, including Cora, Citeseer, Pubmed [16] and Reddit [9].

### 4.1 Experiment Settings

We compare LADIES with the original GCN (full-batch) , GraphSage (neighborhood sampling) and FastGCN (important sampling). We modified the PyTorch implementation of GCN [7] to add our LADIES sampling mechanism. To make the fair comparison only on the sampling part, we also choose the online PyTorch implementation of all these baselines released by their authors and use the same training code for all the methods. By default, we train 5-layer GCNs with hidden state dimension as 256, using the four methods. We choose 5 neighbors to be sampled for GraphSage, 64 and 512 nodes to be sampled for both FastGCN and LADIES per layer. We update the model with a batch size of 512 and ADAM optimizer with a learning rate of 0.001.

For all the methods and datasets, we conduct training for 10 times and take the mean and variance of the evaluation results. Each time we stop training when the validation accuracy doesn't increase a threshold (0.01) for 200 batches, and choose the model with the highest validation accuracy as the convergence point. We use the following metrics to evaluate the effectiveness of sampling methods:

Table 3: Comparison of LADIES with original GCN (Full-Batch), GraphSage (Neighborhood Sampling) and FastGCN (Important Sampling), in terms of accuracy, time, memory and convergence. Training 5-layer GCNs on different node classification datasets (node number is below the dataset name). Results show that LADIES can achieve the best accuracy with lower time and memory cost.

| Dataset | Sample Method | F1-Score(%) | Total Time(s) | Mem(MB) | Batch Time(ms) | Batch Num |
|---|---|---|---|---|---|---|
| Cora (2708) | Full-Batch | $76.5 \pm 1.4$ | $1.19 \pm 0.82$ | 30.72 | $15.75 \pm 0.52$ | $80.8 \pm 51.7$ |
| | GraphSage (5) | $75.2 \pm 1.5$ | $6.77 \pm 4.94$ | 471.39 | $78.42 \pm 0.87$ | $65.2 \pm 52.1$ |
| | FastGCN (64) | $25.1 \pm 8.4$ | $0.55 \pm 0.65$ | 3.13 | $9.22 \pm 0.20$ | $63.2 \pm 71.2$ |
| | FastGCN (512) | $78.0 \pm 2.1$ | $4.70 \pm 1.35$ | 7.33 | $10.08 \pm 0.29$ | $487 \pm 147$ |
| | LADIES (64) | $77.6 \pm 1.4$ | $4.19 \pm 1.16$ | **3.13** | $9.68 \pm 0.48$ | $436 \pm 118.4$ |
| | LADIES (512) | $\mathbf{78.3 \pm 1.6}$ | $\mathbf{0.72 \pm 0.39}$ | 7.35 | $9.77 \pm 0.28$ | $75.6 \pm 37.0$ |
| Citeseer (3327) | Full-Batch | $62.3 \pm 3.1$ | $0.61 \pm 0.70$ | 68.13 | $15.77 \pm 0.58$ | $40.6 \pm 22.8$ |
| | GraphSage (5) | $59.4 \pm 0.9$ | $4.51 \pm 3.68$ | 595.71 | $53.14 \pm 1.90$ | $57.2 \pm 42.1$ |
| | FastGCN (64) | $19.2 \pm 2.7$ | $0.53 \pm 0.48$ | 5.89 | $8.88 \pm 0.40$ | $64.0 \pm 57.0$ |
| | FastGCN (512) | $44.6 \pm 10.8$ | $4.34 \pm 1.73$ | 13.97 | $10.41 \pm 0.51$ | $386 \pm 167$ |
| | FastGCN (1024) | $63.5 \pm 1.8$ | $2.24 \pm 1.01$ | 23.24 | $10.54 \pm 0.27$ | $223 \pm 98.6$ |
| | LADIES (64) | $\mathbf{65.0 \pm .1.4}$ | $2.17 \pm 0.65$ | **5.89** | $9.60 \pm 0.39$ | $232 \pm 66.8$ |
| | LADIES (512) | $64.3 \pm 2.4$ | $\mathbf{0.41 \pm 0.22}$ | 13.92 | $10.32 \pm 0.23$ | $37.6 \pm 11.9$ |
| Pubmed (19717) | Full-Batch | $71.9 \pm 1.9$ | $4.80 \pm 1.53$ | 137.93 | $44.69 \pm 0.57$ | $102 \pm 33.4$ |
| | GraphSage (5) | $70.1 \pm 1.4$ | $5.53 \pm 2.57$ | 453.58 | $44.73 \pm 0.30$ | $74.8 \pm 31.7$ |
| | FastGCN (64) | $38.5 \pm 6.9$ | $0.40 \pm 0.69$ | 1.92 | $7.42 \pm 0.16$ | $58.8 \pm 94.8$ |
| | FastGCN (512) | $39.3 \pm 9.2$ | $\mathbf{0.44 \pm 0.61}$ | 4.53 | $10.06 \pm 0.41$ | $44.8 \pm 55.0$ |
| | FastGCN (8192) | $74.4 \pm 0.8$ | $3.47 \pm 1.16$ | 49.41 | $17.84 \pm 0.33$ | $195 \pm 56.9$ |
| | LADIES (64) | $\mathbf{76.8 \pm 0.8}$ | $2.57 \pm 0.72$ | **1.92** | $9.43 \pm 0.47$ | $277 \pm 82.2$ |
| | LADIES (512) | $75.9 \pm 1.1$ | $2.27 \pm 1.17$ | 4.39 | $10.43 \pm 0.36$ | $245 \pm 84.5$ |
| Reddit (232965) | Full-Batch | $91.6 \pm 1.6$ | $474.3 \pm 84.4$ | 2370.48 | $1564 \pm 3.41$ | $179 \pm 75.5$ |
| | GraphSage (5) | $92.1 \pm 1.1$ | $13.12 \pm 2.84$ | 1234.63 | $121.47 \pm 0.72$ | $81.5 \pm 42.3$ |
| | FastGCN (64) | $27.8 \pm 12.6$ | $2.06 \pm 1.29$ | 3.75 | $7.85 \pm 0.72$ | $57.4 \pm 43.7$ |
| | FastGCN (512) | $17.5 \pm 16.7$ | $\mathbf{0.31 \pm 0.41}$ | 6.91 | $10.01 \pm 0.31$ | $32.1 \pm 72.3$ |
| | FastGCN (8192) | $89.5 \pm 1.2$ | $5.63 \pm 2.12$ | 74.28 | $16.57 \pm 0.58$ | $278 \pm 51.2$ |
| | LADIES (64) | $83.5 \pm 0.9$ | $5.62 \pm 1.58$ | **3.75** | $9.42 \pm 0.48$ | $453 \pm 88.2$ |
| | LADIES (512) | $\mathbf{92.8 \pm 1.6}$ | $6.87 \pm 1.17$ | 7.26 | $10.87 \pm 0.63$ | $393 \pm 74.4$ |

- **Accuracy**: The micro F1-score of the test data at the convergence point. We calculate it using the full-batch version to get the most accurate inference (only care about training).

- **Total Running Time (s)**: The total training time (exclude validation) before convergence point.

- **Memory (MB)**: Total memory costs of model parameters and all hidden representations of a batch.

- **Batch Time and Num**: Time cost to run a batch and the total batch number before convergence.

### 4.2 Experiment Results

As is shown in Table 4, our proposed LADIES can achieve the highest accuracy score among all the methods, using a small sampling number. One surprising thing is that the sampling-based method can achieve higher accuracy than the Full-Batch version, and in some cases using a smaller sampling number can lead to better accuracy (though it may take longer batches to converge). This is probably because the graph data is incomplete and noisy, and the stochastic nature of the sampling method can bring in regularization for training a more robust GCN with better generalization accuracy [10]. Another observation is that no matter the size of the graph, LADIES with a small sample number (64) can still converge well, and sometimes even better than a larger sample number (512). This indicates that LADIES is scalable to training very large and deep GCN while maintaining high accuracy.

As a comparison, FastGCN with 64 and 512 sampled nodes can lead to similar accuracy for small graphs (Cora). But for bigger graphs as Citeseer, Pubmed, and Reddit, it cannot converge to a good point, partly because of the computation graph sparsity issue. For a fair comparison, we choose a higher sampling number for FastGCN on these big graphs. For example, in Reddit, we choose 8192 nodes to be sampled, and FastGCN in such cases can converge to a similar accurate result compared with LADIES, but obviously taking more memory and time cost. GraphSage with 5 nodes to be sampled takes far more memory and time cost because of the redundancy problem we've discussed, and its uniform sampling makes it fail to converge well and fast compared to importance sampling.

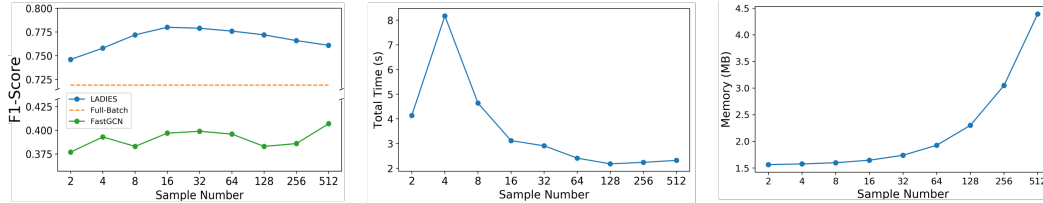

Figure 2: F1-score, total time and memory cost at convergence point for training PubMed, when we choose different sampling numbers of our method. Results show that LADIES can achieve good generalization accuracy (F1-score = 77.6) even with a small sampling number as 16, while FastGCN cannot converge (only reach F1-score = 39.3) with a large sampling number as 512.

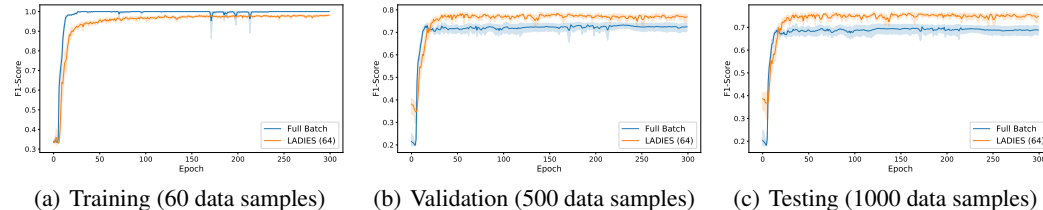

(a) Training (60 data samples)    (b) Validation (500 data samples)    (c) Testing (1000 data samples)

Figure 3: Experiments on the PubMed dataset. We plot the F1-score of both full-batch GCN and LADIES every epoch on (a) Training dataset (b) Validation dataset and (c) Testing dataset.

In addition to the above comparison, we show that our proposed LADIES can converge pretty well with a much fewer sampling number. As is shown in Figure 2, when we choose the sampling number as small as 16, the algorithm can already converge to the best result, with low time and memory cost. This implies that although in Table 4, we choose sample number as 64 and 512 for a fair comparison, but actually, the performance of our method can be further enhanced with a smaller sampling number.

Furthermore, we show that the stochastic nature of LADIES can help to achieve better generalization accuracy than original full-batch GCN. We plot the F1-score of both full-batch GCN and LADIES on the PubMed dataset for 300 epochs without early stop in Figure 3. From Figure 3(a), we can see that, full-batch GCN can achieve higher F1-Score than LADIES on the training set. Nevertheless, on the validation and test datasets, we can see from Figures 3(b) and 3(c) that LADIES can achieve significantly higher F1-Score than full-batch GCN. This suggests that LADIES has better generalization performance than full-batch GCN. The reason is: real graphs are often noisy and incomplete. Full-batch GCN uses the entire graph in the training phase, which can cause overfitting to the noise. In sharp contrast, LADIES employs stochastic sampling to use partial information of the graph and therefore can mitigate the noise of the graph and avoid overfitting to the training data. At a high-level, the sampling scheme adopted in LADIES shares a similar spirit as bagging and bootstrap [1], which is known to improve the generalization performance of machine learning predictors.

## 5 Conclusions

We propose a new algorithm namely LADIES for training deep and large GCNs. The crucial ingredient of our algorithm is layer-dependent importance sampling, which can both ensure dense computation graph as well as avoid drastic expansion of receptive field. Theoretically, we show that LADIES enjoys significantly lower memory cost, time complexity and estimation variance, compared with existing GCN training methods including GraphSAGE and FastGCN. Experimental results demonstrate that LADIES can achieve the best test accuracy with much lower computational time and memory cost on benchmark datasets.

## Acknowledgement

We would like to thank the anonymous reviewers for their helpful comments. D. Zou and Q. Gu were partially supported by the NSF BIGDATA IIS-1855099, NSF CAREER Award IIS-1906169 and Salesforce Deep Learning Research Award. Z. Hu, Y. Wang, S. Jiang and Y. Sun were partially supported by NSF III-1705169, NSF 1937599, NSF CAREER Award 1741634, and Amazon Research Award. We also thank AWS for providing cloud computing credits associated with the NSF BIGDATA award. The views and conclusions contained in this paper are those of the authors and should not be interpreted as representing any funding agencies.

## Footnotes

[2]codes are avaiable at https://github.com/acbull/LADIES

[3]https://zephoria.com/top-15-valuable-facebook-statistics/

[4]We would like to clarify that the proposed layer-dependent importance sampling is different from "layered importance sampling" proposed in [14].

[6]For simplicity, when evaluating the variance we only consider two-layer GCN.

[7] https://github.com/tkipf/pygcn

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
