[Supplementary Material]

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

# A    Complexity comparison between different algorithms

We illustrate how to compute memory and time complexity in this part. The notations remains the same as in table 1 and 2.

The memory requirement for all the mentioned methods consists of two parts, including for storing the intermediate embedding matrices and weight matrices. The first part varies for different algorithms, but the second part is always $\mathcal{O}(LK^2)$ since it has to store $L$ weight matrices with dimension $K \times K$.

For computation cost, it mainly consists of two steps, including the feature propagation and feature transformation. Feature propagation corresponds to the neighbors' embeddings aggregation operation while updating the activations, and the feature transformation corresponds to the linear transformation associated with learnable weight matrix $\mathbf{W}^{(l)}$.

For full-batch GCN, it stores the intermediate embedding matrices for all layers, since it has $L$ layers, each layer has $|\mathcal{V}|$ nodes with dimension of $K$, thus its memory complexity for embedding matrices storage is $\mathcal{O}(L|\mathcal{V}|K)$, combining with weight matrices' memory requirement $\mathcal{O}(LK^2)$, the memory complexity for full-batch GCN is $\mathcal{O}(L|\mathcal{V}|K + LK^2)$ in total. For its time complexity, the feature propagation operation corresponds to sparse-dense matrix multiplication $\mathbf{U} = \mathbf{P}\mathbf{H}^{(l-1)}$, which leads to time complexity $\mathcal{O}(\|\mathbf{A}\|_0 K)$. The feature transformation operation corresponds to dense-dense matrix multiplication $\mathbf{U}\mathbf{W}^{l-1}$, which leads to time complexity $\mathcal{O}(|\mathcal{V}|K^2)$. Thus, the per-batch time complexity for a $L$-layer network is $\mathcal{O}(L\|\mathbf{A}\|_0 K + L|\mathcal{V}|K^2)$ in total.

Now we analyze the complexity of node-wise sampling methods. For GraphSAGE, with batch-size $b$, memory requirement for storing the intermediate embedding matrices is $\mathcal{O}(bKs_{node}^{L-1})$, since it has $\mathcal{O}(bs_{node}^{L-1})$ $K$-dimension embedding aggregation operation. This leads to total memory complexity $\mathcal{O}(bKs_{node}^{L-1} + LK^2)$. For time complexity, for each batch of b nodes, we need to update $\mathcal{O}(s_{node}^{L-1})$ activations for one node. Each new activation needs to aggregate $s_{node}$ embeddings in previous layers, so for one batch, the computation cost for feature propagation is $O(bKs_{node}^L)$. For feature transformation, since each new activation has a vector-matrix multiplication operation after the aggregation, thus the computation cost would be $\mathcal{O}(bK^2s_{node}^{L-1})$ for this part. Therefore, we know the total per-batch time complexity for GraphSAGE is $\mathcal{O}(bKs_{node}^L + bK^2s_{node}^{L-1})$.

For another node-wise sampling strategy VR-GCN, it needs to store all historical activations, so its storage requirement for embedding matrices is $\mathcal{O}(L|\mathcal{V}|K)$, therefore the total memory is $\mathcal{O}(L|\mathcal{V}|K + LK^2)$. The time complexity is the same as GraphSAGE, which is $\mathcal{O}(bDKs_{node}^{L-1} + bK^2s_{node}^{L-1})$.

For layer-wise sampling method FastGCN, for embedding matrices storage, it only has to store $b$ embedding vectors in the last layer, and $(L-1)s_{layer}$ embedding vectors in the previous $L-1$ layers, so the total memory complexity for this part is $\mathcal{O}(bK + (L-1)Ks_{layer})$. Also, it is easy to see that time complexity would be $\mathcal{O}(bKs_{layer} + (L-1)Ks_{layer}^2)$ for feature propagation, and $\mathcal{O}((b+(L-1)s_{layer})K^2)$ for feature transformation. Usually, we have $b < s_{layer}$, so after discarding relatively small terms, the memory complexity in total would be $\mathcal{O}(LKs_{layer} + LK^2)$, and time complexity in total is $\mathcal{O}(LKs_{layer}^2 + LK^2s_{layer})$.

Then, for our proposed LADIES algorithm, since its also a layer-wise sampling method, its memory complexity and time complexity are $\mathcal{O}(LKs_{layer} + LK^2)$ and $\mathcal{O}(LKs_{layer}^2 + LK^2s_{layer})$ respectively, which is same as FastGCN.

# B    Variance comparison between different algorithms

We first provide the following lemma which is useful in our computation of variance.

**Lemma 1** *Given two matrices $\mathbf{A} \in \mathbb{R}^{n \times m}$ and $\mathbf{B} \in \mathbb{R}^{m \times d}$, for any $i \in [m]$ define the probability $p_k = \|\mathbf{A}_{*,k}\|_2^2 / \|\mathbf{A}\|_F^2$, and generate random sketching matrix $\mathbf{S}$ accordingly, it holds that*

$$\mathbb{E}_{\mathbf{S}}[\|\mathbf{A}\mathbf{S}\mathbf{B} - \mathbf{A}\mathbf{B}\|_F^2] = \frac{1}{s}\big(\|\mathbf{A}\|_F^2\|\mathbf{B}\|_F^2 - \|\mathbf{A}\mathbf{B}\|_F^2\big),$$

*where $s$ is the number of samples.*

**Proof** By definition we have

$$\mathbb{E}_\mathbf{S}[\|\mathbf{ASB} - \mathbf{AB}\|_F^2] = \frac{1}{s}\left(\mathbb{E}_k\left[\left\|\mathbf{A}_{*,k}\mathbf{B}_{k,*}/p_k\right\|_F^2\right] - \|\mathbf{AB}\|_F^2\right)$$

$$= \frac{1}{s}\left(\sum_{k=1}^m p_k \cdot \left\|\mathbf{A}_{*,k}\mathbf{B}_{k,*}/p_k\right\|_F^2 - \|\mathbf{AB}\|_F^2\right)$$

$$= \frac{1}{s}\left(\sum_{k=1}^m \frac{1}{p_k} \cdot \|\mathbf{A}_{*,k}\|_2^2\|\mathbf{B}_{k,*}\|_2^2 - \|\mathbf{AB}\|_F^2\right). \qquad (10)$$

Note that $p_k = \|\mathbf{A}_{*,k}\|_2^2/\|\mathbf{A}\|_F^2$, we have

$$\mathbb{E}_\mathbf{S}[\|\mathbf{ASB} - \mathbf{AB}\|_F^2] = \frac{1}{s}\left(\sum_{k=1}^m \|\mathbf{A}\|_F^2\|\mathbf{B}_{k,*}\|_2^2 - \|\mathbf{AB}\|_F^2\right)$$

$$= \frac{1}{s}\left(\|\mathbf{A}\|_F^2\|\mathbf{B}\|_F^2 - \|\mathbf{AB}\|_F^2\right).$$

This completes the proof.

∎

Before computing the variance for each algorithm in detail, we first lay out several definitions. Our goal is to compute the average variance of the approximate embedding for the sampled nodes at the output layer. Specifically, let $\mathcal{S}$ with $|\mathcal{S}| = b$ denote the set of sampled nodes at the output layer, and $\mathbf{Q} \in \mathbb{R}^{b \times |\mathcal{V}|}$ denotes the corresponding row selection matrix that extracts the embedding from the entire embedding matrix. Then the average variance can be defined as $\mathbb{E}[\|\tilde{\mathbf{Z}} - \mathbf{QZ}\|_F]/b$, where $\tilde{\mathbf{Z}} \in \mathbb{R}^{b \times d}$ denotes the approximated intermediate embedding by sampling algorithms and $\mathbf{Z} = \mathbf{PHW}$ denotes the true intermediate embedding. For layer-wise sampling-based methods including FastGCN and LADIES, we sample $bs$ nodes at the hidden layer. For node-wise sampling-based methods including GraphSAGE and VR-GCN, we sample $m$ neighbors for each node.

In order to explicitly compare the variance of different algorithms, we made the following assumptions on the Laplacian matrix $\mathbf{P}$.

**Assumption 1** *We assume there exist absolute constants $C_1$ and $C_2$ such that the following holds for all $i \in [|\mathcal{V}|]$,*

$$\|\mathbf{P}_{i,*}\|_0 \leq \frac{C_1}{|\mathcal{V}|}\sum_{i=1}^{|\mathcal{V}|}\|\mathbf{P}_{i,*}\|_0 \quad and \quad \|\mathbf{P}_{i,*}\|_2^2 \leq \frac{C_2}{|\mathcal{V}|}\sum_{i=1}^n\|\mathbf{P}_{i,*}\|_F^2.$$

Then we make the following assumption on the matrix product $\mathbf{HW}$.

**Assumption 2** *We assume that the $\ell_2$-norm of each row of $\mathbf{HW}$ is upper bounded by a constant, i.e., there exists a constant $\phi$ such that $\|\mathbf{H}_{i,*}\mathbf{W}\|_2 \leq \phi$ for all $i \in [|\mathcal{V}|]$.*

**Variance of FastGCN.** Note that FastGCN does not apply layer-dependent sampling and focus on the variance $\mathbb{E}[\|\mathbf{PSHW} - \mathbf{Z}\|_F^2]$, where $\mathbf{S}$ is generated according to (5). By applying the importance probabilities configured in [4], i.e., $p_i = \|\mathbf{P}_{*,i}\|_2/\|\mathbf{P}\|_F$, it can be derived that

$$\mathbb{E}[\|\tilde{\mathbf{Z}} - \mathbf{QZ}\|_F^2] = \frac{b}{|\mathcal{V}|}\mathbb{E}[\|\mathbf{PSHW} - \mathbf{Z}\|_F^2]$$

$$= \frac{b}{|\mathcal{V}|s}\left(\sum_{k=1}^{|\mathcal{V}|}\frac{1}{p_k}\cdot\left\|\mathbf{P}_{*,k}\right\|_2^2\|\mathbf{H}_{k,*}\mathbf{W}\|_2^2 - \|\mathbf{Z}\|_F^2\right)$$

$$= \frac{b\left(\|\mathbf{P}\|_F^2\|\mathbf{HW}\|_F^2 - \|\mathbf{Z}\|_F^2\right)}{|\mathcal{V}|s}.$$

where the first equation holds due to the fact that $\mathbb{E}[\mathbf{Q}] = b\mathbf{I}/|\mathcal{V}|$ and $\mathbf{Q}$ and $\mathbf{S}$ are independent, and the second equality is by Lemma 1. Then under Assumption 2, we have $\|\mathbf{HW}\|_F^2 \le |\mathcal{V}|\phi$, which further implies that

$$\mathbb{E}[\|\tilde{\mathbf{Z}} - \mathbf{QZ}\|_F^2]/b \le \frac{\phi\|\mathbf{P}\|_F^2}{s}.$$

**Variance of GraphSAGE.** We assume for each node, GraphSAGE randomly sample $m$ nodes from its neighbors to estimate the embedding. Also, the randomness of sampling at the hidden layer is independent of that at the output layer. Therefore, it is clearly that

$$
\begin{aligned}
\mathbb{E}[\|\tilde{\mathbf{Z}} - \mathbf{QZ}\|_F^2] &= \frac{b}{|\mathcal{V}|} \cdot \sum_{i=1}^{|\mathcal{V}|} \mathbb{E}[\|\tilde{\mathbf{Z}}_{i,*} - \mathbf{Z}_{i,*}\|_2^2] \\
&= \frac{b}{|\mathcal{V}|} \cdot \sum_{i=1}^{|\mathcal{V}|} \frac{\|\mathbf{P}_{i,*}\|_0}{m} \left( \sum_{j=1}^{|\mathcal{V}|} \|\mathbf{P}_{i,j}\mathbf{H}_{j,*}\mathbf{W}\|_2^2 - \|\mathbf{P}_{i,*}\mathbf{HW}\|_F^2 \right) \\
&= \frac{b}{m|\mathcal{V}|} \sum_{i=1}^{|\mathcal{V}|} \|\mathbf{P}_{i,*}\|_0 \left( \sum_{j=1}^{|\mathcal{V}|} \|\mathbf{P}_{i,j}\mathbf{H}_{j,*}\mathbf{W}\|_2^2 - \frac{b}{m|\mathcal{V}|} \|\mathbf{PHW}\|_F^2 \right).
\end{aligned}
\tag{11}
$$

Under Assumption 2 it is clear that

$$
\begin{aligned}
\mathbb{E}[\|\tilde{\mathbf{Z}} - \mathbf{QZ}\|_F^2]/b &\le \frac{1}{m|\mathcal{V}|} \sum_{i=1}^{|\mathcal{V}|} \|\mathbf{P}_{i,*}\|_0 \sum_{j=1}^{|\mathcal{V}|} \|\mathbf{P}_{i,j}\mathbf{H}_{j,*}\mathbf{W}\|_2^2 \\
&\le \frac{C_1 D}{m|\mathcal{V}|} \sum_{i=1}^{|\mathcal{V}|} \sum_{j=1}^{|\mathcal{V}|} \|\mathbf{P}_{i,j}\mathbf{H}_{j,*}\mathbf{W}\|_2^2 \\
&\le \frac{C_1 D \phi \|\mathbf{P}\|_F^2}{m|\mathcal{V}|},
\end{aligned}
$$

where the second inequality is by Assumption 1 and the definition of $D$, and the second inequality is by Assumption 2.

**Variance of VR-GCN.** Assuming we have history activation matrix $\bar{\mathbf{H}}$, VR-GCN turns the estimation of $\mathbf{P}_{i,*}\mathbf{HW}$ to that of $\mathbf{P}_{i,*}(\mathbf{H} - \bar{\mathbf{H}})\mathbf{W}$. Thus, we can simply derive the variance of VR-GCN by replacing $\mathbf{H}$ in (11) with $\mathbf{H} - \bar{\mathbf{H}}$. Therefore, the variance of VR-GCN is

$$\mathbb{E}[\|\tilde{\mathbf{Z}} - \mathbf{QZ}\|_F^2] = \frac{b}{m|\mathcal{V}|} \sum_{i=1}^{|\mathcal{V}|} \|\mathbf{P}_{i,*}\|_0 \left( \sum_{j=1}^{|\mathcal{V}|} \|\mathbf{P}_{i,j}(\mathbf{H}_{j,*} - \bar{\mathbf{H}}_{j,*})\mathbf{W}\|_2^2 - \frac{b}{m|\mathcal{V}|} \|\mathbf{P}(\mathbf{H} - \bar{\mathbf{H}})\mathbf{W}\|_F^2 \right).$$

The variance of VR-GCN no longer depends on the norm of $\|\mathbf{H}_{j,*}\mathbf{W}\|_2^2$ but $\|(\mathbf{H}_{j,*} - \bar{\mathbf{H}}_{j,*})\mathbf{W}\|_2^2$. We define $\Delta\phi = \max_j \|(\mathbf{H}_{j,*} - \bar{\mathbf{H}}_{j,*})\mathbf{W}\|_2^2$ and thus

$$\mathbb{E}[\|\tilde{\mathbf{Z}} - \mathbf{QZ}\|_F^2]/b \le \frac{D\Delta\phi\|\mathbf{P}\|_F^2}{m|\mathcal{V}|}.$$

**Variance of LADIES.** Different from other algorithms, the random samples at the output layer and hidden layer are not independent. We first define $\mathcal{V}(\mathcal{S})$ by the union of neighbors of nodes in $\mathcal{S}$ (i.e., nodes sampled in the upper layer). Then we have

$$
\begin{aligned}
\mathbb{E}[\|\tilde{\mathbf{Z}} - \mathbf{QZ}\|_F^2] &= \mathbb{E}\big[\mathbb{E}[\|\tilde{\mathbf{Z}} - \mathbf{QZ}\|_F^2|\mathbf{Q}]\big] \\
&= \mathbb{E}\big[\mathbb{E}_\mathbf{S}[\|\mathbf{QPSLHW} - \mathbf{QPLHW}\|_F^2]|\mathbf{Q}\big] \\
&= \frac{1}{s}\mathbb{E}\big[\|\mathbf{QP}\|_F^2\|\mathbf{LHW}\|_F^2 - \|\mathbf{QZ}\|_F^2\big],
\end{aligned}
$$

where $\mathbf{L}$ is a diagonal matrix with $\mathbf{L}_{i,j} = 1$ if $i = j, i \in \mathcal{V}(\mathcal{S})$ and $\mathbf{L}_{i,j} = 0$ otherwise, and the second equation is by Lemma 1 and our choice of importance probabilities.

| (a) Sample Number=1 | (b) Sample Number=2 | (c) Sample Number=8 |
| (d) Sample Number=32 | (e) Sample Number=128 | (f) Sample Number=512 |

Figure 4: Training Curve of Pubmed with different sampling numbers.

Figure 5: Convergence Curve of Pubmed with different sampling methods.

Note that by Assumption 1, we have

$$\|\mathbf{Q}\mathbf{P}\|_F^2 = \sum_{i \in \mathcal{S}} \|\mathbf{P}_{i,*}\|_2^2 \le \frac{C_2 b \|\mathbf{P}\|_F^2}{|\mathcal{V}|}.$$

Then under Assumption 2, we have

$$\mathbb{E}[\|\tilde{\mathbf{Z}} - \mathbf{Q}\mathbf{Z}\|_F^2]/b \le \frac{1}{sb} \mathbb{E}[\|\mathbf{Q}\mathbf{P}\|_F^2 \cdot \|\mathbf{L}\mathbf{H}\mathbf{W}\|_F^2] \le \frac{C_2 \phi \|\mathbf{P}\|_F^2}{|\mathcal{V}|s} \mathbb{E}[|\mathcal{V}(\mathcal{S})|] = \frac{C_2 \phi \|\mathbf{P}\|_F^2}{|\mathcal{V}|s} \bar{V}(b).$$

where $\bar{V}(b)$ denotes the average number of nodes that are connected to the nodes sampled in the training batch.

## C  Convergence Curves

Figure 4 shows the training curve of LADIES on Pubmed, with different sampling number. We can see that even with a very small sampling number (i.e., 8), it's already converged pretty well. On the contrary, our previous experiments show that FastGCN with a huge sampling number (i.e., 512) cannot even converge. Figure 5 shows that our method can achieve better and robust convergence performance than FastGCN with smaller sampling number and fewer time.

## D  Differences with FastGCN

Since our proposed LADIES also uses Layer-wise Importance Sampling, here we emphasize the key differences in the following aspects: (1) our algorithm restricts the candidate nodes in the union of the neighborhoods of the sampled nodes in the upper layer, which can lead to a much denser sampled adjacency matrix compared with FastGCN; (2) our sampling method is layer-dependent, and the importance sampling probabilities are novelly derived from optimizing on the theoretical derivation of the modified Laplacian matrix (see formula (8)), which leads to smaller estimation variance than FastGCN, as shown in Table 2; and (3) our algorithm uses normalization to the modified Laplacian matrix in each layer, which further stabilizes the forward process and avoids exploding/vanishing gradient. Because of the above innovative and principled algorithmic designs, our algorithm enjoys lower time and memory complexities, as well as better predictive performance than FastGCN in both theory and practice.