[Reviews · NeurIPS 2019]

Reviewer 1



* Originality: the method is novel. The idea of sampling based on upper layers is itself interesting. * Quality: the work is based on solid arguments and theoretical proofs. The experiments are thoughtful, although lack analysis. * Clarity: the paper is in general easy to read (thought I didn't check the maths carefully). - Table 3: Isn't that LADIES should be an approximation of original GCN? If that's true, why is Full-Batch outperformed (in terms of F1) by LADIES? * significance: this work will have a high impact because of its simplicity, solidity, and high performances.

Reviewer 2



The topic of the paper is interesting. The paper seems technically sound and generally well-written. However, it is quite difficult to read in my opinion. I have only one suggestion for avoiding confusion. - Similar names can generate several misunderstandings and confusion. The name/title "Layered Importance Sampling" recall me the following Monte Carlo sampling approach, L. Martino, V. Elvira, D. Luengo, J. Corander, "Layered Adaptive Importance Sampling", Statistics and Computing, Volume 27, Issue 3, Pages: 599-623, 2017. I think you should clarify that your framework is different. More generally, "importance sampling" in the Monte Carlo community has a clear different meaning (referring to a specific algorithm).

Reviewer 3



This paper focuses on solving the sparsity issue in the importance sampling of FastGCN and propose a layer-dependent importance sampling schema. However, the main modification is sampling the neighbors based on a defined probability from the union neighborhood of nodes in the upper layer. However, this is a small modification of FastGCN since for a mini-batch training, it is straightforward to improve the sampling efficiency by limiting the candidate nodes to the neighborhood of the nodes in the upper layer. In fact, this is just a trick in the coding implementation of FastGCN in terms of this point claimed by the authors, and cannot be considered as a novel improvement in the level of NeurIPS although you have conducted the corresponding experiments.

[Author Response · NeurIPS 2019]

**To Reviewer #1**: Thanks for your positive comments on our paper!

**Q1: Why is Full-Batch outperformed by LADIES?**

**A1:** It is true that LADIES is designed as an approximation of original GCN. To answer your question, here we plot the F1-score of both full-batch GCN and LADIES on the PubMed dataset for 300 epochs without early stop in Figure 1. From Figure 1(a), we can see that, full-batch GCN can achieve higher F1-Score than LADIES on the training set. Nevertheless, on the validation and test datasets, we can see from Figures 1(b) and 1(c) that LADIES can achieve significantly higher F1-Score than full-batch GCN. This suggests that LADIES has better generalization performance than full-batch GCN. The reason is: real graphs are often noisy and incomplete. Full-batch GCN uses the entire graph in the training phase, which can cause overfitting to the noise. In sharp contrast, LADIES employs stochastic sampling to use partial information of the graph, and therefore can mitigate the noise of graph and avoid overfitting to the training data. At a high-level, the sampling scheme adopted in LADIES shares a similar spirit as bagging/bootstrap [1], which is known to improve the generalization performance of machine learning predictors. We will add these additional plots as well as the above explanation in the camera-ready version.

[1] Leo Breiman. Bagging predictors. Machine learning, 24 (2):123–140, 1996

(a) Training (60 data samples)    (b) Validation (500 data samples)    (c) Testing (1000 data samples)

Figure 1: Experiments on the PubMed dataset, which contains 19717 nodes and 44338 edges. We plot the F1-score of both full-batch GCN and LADIES every epoch on (a) Training dataset (b) Validation dataset and (c) Testing dataset.

**To Reviewer #2**

We would like to highlight the major contributions of our paper as follows: how to efficiently train GCN on large-scale graphs is a long-standing research problem since the seminal work of GCN (Kipf et al., 2017). Our proposed algorithm can use a remarkably small sampled node number (64 nodes per GCN layer, which determines the total memory consumption) to achieve the best known performance with small time and memory consumption, compared with existing state-of-the-art GCN training algorithms. This suggests that using our algorithm, we can train an accurate GCN for graphs with very large scale and high density.

**Q1: "Similar names generate several misunderstandings and confusions"**

**A1:** We apologize the title causes confusing. However, we would like to clarify that our paper entitles "Layer-dependent importance sampling" rather than "Layered importance sampling". We would like to emphasize that our "Layer-dependent importance sampling" is fundamentally different from "Layered importance sampling" discussed in the paper pointed out by you. We will clarify it in future version of the paper.

**Q2: "Importance sampling in the Monte Carlo community has a clear different meaning"**

**A2:** We would like to clarify that the terminology of "importance sampling" for GCN training follows the existing work [2, 3]. Moreover, the "importance sampling" used in our paper belongs to the general category of importance sampling. As we stated in the paper, we sample a subset of nodes to approximate the exact computation of embeddings in each layer. In order to reduce the estimation variance, we assign important weights to all candidate nodes (as computed in (8)) and then apply non-uniformly sampling according to the weights to estimate the embeddings.

**To Reviewer #3**

**Q1: "The proposed algorithm cannot be considered as a novel improvement in the level of NeurIPS ..."**

**A1:** We respectfully disagree with your comment that our algorithm is just a trick in the implementation of FastGCN. Compared with FastGCN, our algorithm is different in the following aspects: (1) our algorithm restricts the candidate nodes in the union of the neighborhoods of the sampled nodes in the upper layer, which can lead to a much denser sampled adjacency matrix compared with FastGCN; (2) our sampling method is layer-dependent, and the importance sampling probabilities are novelly derived from optimizing on the theoretical derivation of the modified Laplacian matrix (see (8)), which leads to smaller estimation variance than FastGCN, as shown in Table 2; and (3) our algorithm uses normalization to the modified Laplacian matrix in each layer, which further stabilizes the forward process and avoids exploding/vanishing gradient. Because of the above innovative and principled algorithmic designs, our algorithm enjoys lower time and memory complexities, as well as better predictive performance than FastGCN in both theory and practice. We believe the improvement of the proposed algorithm is novel and meets the standard of NeurIPS.

[Meta-Review · NeurIPS 2019]

All reviewers agreed that this work addresses a highly relevant topic, and that the theoretical results regarding the variance reduction is certainly interesting. There was, on the other hand, some disagreement about the "true" novelty and the significance of the differences with respect to related approaches, which could not be fully resolved in the discussions following the author rebuttal -- but such a controversial discussion might also be seen as an indicator for the overall importance of this paper.